# Self-Reported Pesticide Exposure During Pregnancy and Pesticide-Handling Knowledge Among Small-Scale Horticulture Women Workers in Tanzania, a Descriptive Cross-Sectional Study

**DOI:** 10.3390/ijerph22010040

**Published:** 2024-12-30

**Authors:** William Nelson Mwakalasya, Simon Henry Mamuya, Bente Elisabeth Moen, Aiwerasia Vera Ngowi

**Affiliations:** 1Department of Environmental and Occupational Health, School of Public Health and Social Sciences, Muhimbili University of Health and Allied Sciences, Dar es Salaam P.O. Box 65001, Tanzania; willykitwi@yahoo.com (W.N.M.); mamuyasimon2@gmail.com (S.H.M.); vera.ngowi@gmail.com (A.V.N.); 2Department of Global Public Health and Primary Care, Centre for International Health, University of Bergen, 5020 Bergen, Norway

**Keywords:** pesticides exposure, pregnant women, pesticide label information, pesticide handling, horticulture

## Abstract

Women constitute most of the global horticulture workforce, where pesticide use is prevalent. Protecting their health, particularly during pregnancy, is essential. However, knowledge about practices among pregnant employees that cause exposure to pesticides is limited. This study aims to identify such practices and assess the impact of pesticide-handling knowledge on exposure. A cross-sectional survey was conducted among 432 small-scale horticulture women workers in Tanzania from October 2022 to April 2023. The women were interviewed using a self-report questionnaire, with descriptive statistics, Pearson’s chi-square tests, and T-tests used for data analysis. In total, 86% of participants worked in horticulture during pregnancy, with 47.5% continuing into the third trimester. Many engaged in weeding within 24 h of spraying (58.4%) and washing pesticide-contaminated clothes (51.7%). Most of the women (93.1%) had limited knowledge of pesticide handling, though some understood mixing (62.5%) and spraying (64.1%) instructions on labels. This study suggests that women working in horticulture are exposed to pesticides during pregnancy partly due to limited knowledge of safe pesticide handling. These exposures are largely shaped by the working conditions, which may place both pregnant women and their offspring at risk of hazardous pesticide exposure. Hence, there is a need for guidelines and policies towards protecting women working in agriculture.

## 1. Introduction

Around 40% of the global agricultural workforce consists of women, the majority being from low- and middle-income countries. In sub-Saharan Africa, over 60% of all working women are in agriculture [1]. The population-weighted average from six African countries (Malawi, Tanzania, Uganda, Nigeria, Niger, and Ethiopia) estimated the female labor share in crop production to be around 40% [2]. In Tanzania, 67% of the total female labor force works in the agriculture sector [3]. Horticulture, the fastest-growing sector of agriculture in the world, contributes 33% of the total agricultural output [4], and women are estimated to be involved in up to 70% of the actual farm work [5]. In Tanzania, women account for approximately 65 to 70% of the labor force in horticulture farming [6]. Thus, women constitute a significant workforce in the horticulture sector.

Management of horticulture crops often involves the intensive use of pesticides—chemicals or substances used to control, repel, or eliminate pests that can harm crops. While pesticides are valuable for protecting crops, they can have serious adverse health effects on humans. Existing scientific evidence suggests that pesticide exposure can have adverse health effects on the respiratory system, nervous system, skin, and eyes of agricultural workers, nearby communities, and consumers, resulting in endocrine, reproductive, carcinogenic, neurological, and other disorders [7,8]. Exposure of pregnant women to pesticides might cause adverse health effects for the fetus because some pesticides can cross the placental barrier [9]. Adverse pregnancy outcomes, such as impaired fetal functional immunity [10], increased risk of congenital anomalies [11], impaired neuropsychological development [12], preterm birth, small size for gestational age, and low birth weight [13], have been suggested to be caused by prenatal pesticide exposure.

Over a decade ago, a study assessed the levels of organochlorine pesticide metabolites in maternal blood samples obtained from healthy pregnant women in northern Tanzania. The study revealed the presence of organochlorine pesticide metabolites in all maternal blood samples [14]. Subsequently, a study conducted in the southern region of the country, focusing on women working in horticulture, revealed that children born to mothers exposed to pesticides exhibited impaired neurodevelopment [15]. These localized findings shed some light on the likelihood of pesticide exposure during the critical period of pregnancy, yet the specific practices associated with such exposures in the course of work remain unclear. A good understanding of these practices is crucial for further research on this topic.

The present study aimed to identify pesticide-handling practices and knowledge related to pesticide handling during pregnancy among women working in horticulture. Unfolding these practices is essential for future studies and the implementation of interventions to protect pregnant women and their unborn children. This topic aligns with the pursuit of Sustainable Development Goal 3.9, which aims at reducing the number of fatalities and diseases caused by hazardous chemicals [16].

## 2. Materials and Methods

### 2.1. Study Design and Population

We conducted a descriptive cross-sectional study in three regions of Tanzania: Pwani, Morogoro, and Mbeya. One district was selected in each region: Bagamoyo, Mvomero, and Mbarali, respectively. The districts were chosen for their high engagement in horticulture. In each district, the two wards with the most households involved in horticulture were selected with the assistance of District Agriculture Officers (DAOs) (Figure 1). In Mvomero, however, the study was conducted in only one ward (Nyandira) due to the sudden departure of a trained research assistant for family reasons, leaving insufficient time to train a replacement.

A priori power analysis was conducted using G*Power version 3.1.9.7 [17] to estimate the sample size, drawing on data from a study that examined self-reported pesticide exposure during pregnancy (N = 5997) [18]. The study revealed that 12.6% of the pregnant women reported using pesticides during the 1st and 3rd trimesters. Given a Type I error rate (α) of 0.05 and a power of 0.80, a minimum sample size of 391 was determined for this effect size. Therefore, the obtained sample size of 432 was sufficient for this study. The sample size was distributed to the three study areas depending on the size of land available for horticulture in the area.

In collaboration with Ward Agriculture Extension Officers (WAEOs), areas actively engaged in horticulture were identified, focusing on those with larger farmlands. Hamlet leaders were introduced to the study and asked to identify households with women meeting the inclusion criteria: those working in horticulture, having a child aged 4–6 years, and who were employed in horticulture before conception and during pregnancy. From the identified households, every third household was selected. The interviews were conducted by trained research assistants at home or on the farm using a digital questionnaire and uploaded to the Kobo Toolbox database daily for secure storage.

### 2.2. Background Information and Exposure Assessment

The questionnaires guided the women in recalling instances of pesticide exposure that occurred during pregnancy, providing researchers with information not documented in any farm or health facility registers. The use of self-reported exposure has been validated and proven to yield consistent findings [19] and accurate recall [20,21]. The questionnaire was divided into two sections. The first section collected the women’s background information, including age, distance from the house to the horticulture farm, number of years living in the area, number of years in horticulture, and number of years in horticulture before their pregnancy. The second section included questions about pesticide handling, and these were adapted from a questionnaire used in a PhD thesis on farmers and pesticide exposure from South Africa [22]. The section had questions on activities practiced during pregnancy which are known to involve direct contact with pesticides including mixing, spraying, weeding, and harvesting within 24 h after pesticide spraying, washing clothes used when applying pesticides, cleaning spray equipment, burning empty pesticide containers, reusing empty pesticide containers, and eating sprayed vegetables within 24 h after spraying. The questions had four options (Never, I don’t remember, Sometimes, and Often), and after each question, all the women who responded “Sometimes” or “Often” had a follow-up question about gestation age.

### 2.3. Knowledge Assessment

Knowledge of safe pesticide handling and use was assessed by two questions. The questions focused on assessing the mothers’ ability to comprehend pesticide information displayed on the packaging label of the pesticide used. Mothers were initially asked whether they knew that pesticide-handling information and information about how to use the pesticide were displayed on the label. Then, mothers were prompted with an open-ended question to recall and list all pertinent information communicated by the packaging label.

The list of items given by participants during the interviews included eight (8) main items: mixing, spraying, first aid, toxicity, personal protection, pesticide name, storage, and disposal. Each item mentioned was given a ‘yes’, and a ‘no’ was given to an item for participants who did not mention the item. Later, the answers were scored. Scores of 1 and 0 were assigned for ‘yes’ and ‘no’, respectively. Hence, the score ranged from 0 (minimum) to 8 (maximum). We adopted the score categorization from Monger et al., 2023 [23]. A cutoff level of ≤50% (≤4 points) was considered low, and >50% (5 or more points) was regarded as high knowledge.

### 2.4. Data Analysis

The data collected was downloaded, cleaned, and imported into IBM SPSS Version 23. Frequencies and percentages were used to summarize the sociodemographic variables. Pesticide exposure practices (dependent variables) were first dichotomized from a 4-point scale into a binary variable [24]. “Never” was coded as 0 or “did not practice”, “Sometimes” and “Often” were coded as 1 or “practiced”, and “I don’t remember” was coded as a missing variable. Chi-square and Fisher’s exact tests were used to compare the distribution of sociodemographic and exposure variables among the three study areas. A *p*-value of ≤0.05 was used as a measure of statistical significance.

## 3. Results

### 3.1. Participant Characteristics

A total of 432 women (99%) consented out of 436 who were asked to participate. Four women did not want to participate because they were not satisfied with the expected benefits of the study. The mean age of participants was about 34 years, with most (55.1%) aged 31–40 years. The women from Mbarali were significantly younger (*p* < 0.001) compared to those from the other study areas. The chi-square test of independence showed that participants from the three study areas also differed in terms of years involved in horticulture work (*p* < 0.001). Most of the women had worked in horticulture for less than five years (55.3%), but more than 40% had worked there for more than six years before their recent pregnancy. Only 10.6% of the women decided to stop horticulture work in the first trimester; most continued working until the second (34.7%) or third (41.0%) trimesters of pregnancy (Table 1).

### 3.2. Pesticide Exposure During Pregnancy

In total, 373 women had been at work in horticulture during their pregnancy (Table 1). These women were asked about their involvement in activities that result in exposure to pesticides during pregnancy. Despite their awareness of being pregnant, some of these women had participated in activities that involved direct contact with pesticides (Table 2). For instance, 39.5% of the surveyed women indicated involvement in pesticide spraying during pregnancy. In addition, a considerable proportion of the women reported re-entry into the fields within 24 h of spraying, with 59.6% engaging in weeding and 23.1% participating in harvesting during pregnancy. Furthermore, 45.8% of the women acknowledged consuming horticultural crops within 24 h of pesticide spraying during pregnancy. A chi-square test of independence and Fisher’s exact test showed that there was an association between the study area and women’s engagement in activities involving direct contact with pesticides (*p* ≤ 0.05) (Table 2).

As shown earlier, only 10.6% of the women decided to stop horticulture work due to pregnancy within the first trimester (Table 1). More than half of those who continued working engaged in weeding throughout the first (52.0%) and second (53.1%) trimesters, with only a few of them being able to continue with the task in the third trimester. Pesticide spraying was also a common task that more than one-third of the women reported continuing to do in the second and third trimesters of pregnancy. Moreover, washing clothes used during pesticide spraying was a first-trimester task for 46.9% of the women who continued working during pregnancy (Table 3).

### 3.3. Knowledge of Pesticide Handling Among Women in Horticulture Work

When the women were asked if they were aware of the presence of important pesticide information displayed on the packaging label, approximately 70% of them (*n* = 301) reported being aware, and among these, 90.4% (*n* = 272) had low knowledge. Regarding the women’s knowledge of safe pesticide handling, a substantial majority, 93.3% (*n* = 403), of the women had low knowledge. The results of a chi-square test of independence showed that knowledge did not have any significant association with years spent in horticulture (experience) (*p* = 0.688) or the mother’s age (*p* = 0.558). The level of knowledge was associated with the study areas (*p* < 0.001), with Bagamoyo having the highest proportion of women with high knowledge. Also, there was a significant association between knowledge and gestation age when the pregnant women decided to stop horticulture work (*p* = 0.001) (Table 4).

A summary of the frequency distribution for each category of label information mentioned by the women is provided in Table 5. Most of the women were aware that pesticide labels include directives for spraying (64.1%) and mixing (62.5%). However, a smaller percentage of the women knew that the labels contain information on toxicity, disposal, storage, first aid, and personal protection.

## 4. Discussion

In this study, we identified practices which predispose pregnant women working in horticulture to pesticide exposure. Our results indicate that women working on small-scale horticulture farms continue engaging in activities that involve direct contact with pesticides during pregnancy, exposing themselves and the fetus to hazardous pesticides. Common horticulture activities practiced by pregnant women include spraying pesticides, weeding and harvesting within 24 h of spraying, washing clothes used for pesticide spraying, and consuming horticulture crops within 24 h of spraying. The proportion of women practicing these activities differed across study areas and trimesters. Just over a tenth of the women decided to stop early, and about half of those who continued made it to the third trimester of pregnancy. We additionally found that the practices might have been influenced by knowledge of pesticide handling because most of the women in our study had low knowledge.

Despite the prevailing social construct that associates pesticide handling and application with men’s roles [25,26], this study presents evidence of women’s involvement in these tasks. However, the proportion of women engaged in spray work is notably lower than that of men. For instance, on horticulture farms in Kenya, only 11.5% of women participated in pesticide spraying [27], which is lower than the percentage of women who not only sprayed pesticides but also sprayed pesticides during pregnancy reported in this study. In contrast, a recent study conducted in Thailand involving 78 mothers with children aged 0 to 72 months found that 70.5% of the mothers participated in pesticide spraying [28]. In both studies, most of the women were disproportionately subjected to pesticide exposure through other tasks on the farm, such as planting, weeding, and harvesting, which is similar to the findings of this study. Exposure to pesticides during field activities is considered high due to their occurrence in recently sprayed fields [29], and workers do not consider them to be risky [30]. This is why women usually perform re-entry activities without adequate protection [31]. Interestingly, a study conducted within the vegetable farming community in Ghana did not identify a statistically significant association between engaging in re-entry activities and alterations in serum concentrations of cholinesterase [32], suggesting that farm tasks alone might not be enough in establishing pesticide exposure.

Occupational exposure to pesticides during pregnancy is largely understudied, as most available studies tend to focus on domestic and environmental sources of exposure. Considering that half of the global labor force is employed in agriculture [33], it becomes evident that occupational exposure in this sector is a critical area requiring examination. We found that some women engaged in activities involving exposure to pesticides until the third trimester of pregnancy, mirroring findings from Thailand [34], where nearly half of the women continued working in agriculture, and more than a quarter applied pesticides during pregnancy. The findings of this study also support what has been reported by Maritano et al. (2022), who studied pregnant women from the Italian Nascita e Infanzia: gli Effetti dell’Ambiente (NINFEA) birth cohort, with the proportion of women engaged in agricultural activities decreasing from 21.3% in the first trimester to 13.7% in the third trimester [18]. In contrast to the findings for the women from the NINFEA cohort, where only 4.7% of the women reported using pesticides in the third trimester, in this study, more than one-third of the women reported using pesticides in the third trimester. Although hospital-based studies report small proportions of women involved in occupations that expose them to pesticides during pregnancy [35], they also report high proportions from domestic exposures [36,37,38]. Therefore, these reports suggest that pesticide exposure throughout pregnancy is common and not limited to occupational exposure. Thus, protective interventions are necessary both at home and at work.

It was observed in this study that most of the women have low knowledge of pesticide handling. Only a few of the women knew that the pesticide packaging label displayed first aid, toxicity, and personal protection information. The level of knowledge in this study was lower than that of previous studies [39,40], possibly because of its unique focus on women working on small-scale horticulture farms and its approach of using information on pesticide package labels, which is often ignored by pesticide users [41]. However, these results agree with the findings reported by Pandiyan et al. (2023), who demonstrated that most of the farm workers who attended a tertiary cancer care hospital in Hyderabad, India, did not have any knowledge of the toxicity signs displayed on the pesticide container [38]. Generally, male farmers are reported to have better knowledge of pesticide use and handling than females [38], possibly because literacy rates are either higher for males than for females [41] or because most pesticide-related educational programs are not gender sensitive, as they are biased toward men [4]. Evidence shows that educational programs have a significant impact on the knowledge of pesticide safety practices [42,43]; hence, gender-inclusive educational initiatives should be introduced to enhance women’s understanding and awareness of pesticide safety.

In its Safety and Health in Agriculture Convention (No. 184), adopted in 2001, the International Labor Organization (ILO) highlighted the dangerous nature of agricultural employment. The convention calls for national agricultural safety and health policies to eliminate, minimize, or regulate dangers in the agricultural working environment to prevent accidents and injuries to health deriving from, related to, or happening during work. In particular, special provisions have been made for women working in the sector in Article 18, which states that “measures shall be taken to ensure that the special needs of women agricultural workers are taken into account in relation to pregnancy, breastfeeding and reproductive health” [44]. The convention entered into force in September 2003, but after 20 years of existence, only 21 countries have ratified it. This undermines efforts toward safeguarding women working in the agricultural sector.

Our study had some limitations. The study had a cross-sectional design. We only collected data at a single point in time, offering a glimpse of a population’s features or connections between variables. This design did not provide an opportunity to monitor the study population over time and did not monitor the women during their pregnancies. To obtain information about their work with pesticides, we had to rely on the memory of the interviewed women. This may have led to over-reporting as well as under-reporting of work with pesticides during pregnancy, and the size of this problem is not known to us. Hopefully, the recall bias was low, as many of the practices under investigation were not isolated events but rather ongoing activities, making them more memorable. Also, this was the only method available for obtaining the information, as there were no records from the companies or in any database of authorities with information about pesticide exposure and work. There was a notable disparity in the sample size between the Bagamoyo and Mbarali districts in comparison to the Mvomero district. The smaller sample size in Mbarali can be attributed to the fact that a significant proportion of women in this region regarded horticulture as a secondary pursuit, prioritizing rice farming. Consequently, the number of women who met the predefined inclusion criteria was lower than initially anticipated, but this did not affect the outcome of the study. In the case of Bagamoyo, data collection coincided with a dry season, during which the water sources required for the irrigation of horticulture crops had dried. Hence, many women transitioned to alternative activities, rendering them less accessible during the data collection. Despite these differences, the findings are consistent across all three areas, suggesting that the findings are robust and valid.

## 5. Conclusions

The data from this study suggest that women in horticulture perform activities during their work that may cause pesticide exposure, and that this exposure may occur also during pregnancy. Many of the women have limited knowledge of pesticide handling. The women reported being involved in spraying pesticide, re-entry a few hours after pesticide application for weeding and harvesting, and washing clothes and equipment used during spraying. Some of the women reported eating horticulture crops sprayed with pesticides within 24 h. All these practices may cause pesticide exposure of pregnant women and their offspring. Unfortunately, no serious local or international efforts are directed toward protecting women in agricultural workplaces. The findings from this study should serve as a wake-up call for all responsible parties to spearhead policy dialogues in line with the ILO Convention No. 184.

## Figures and Tables

**Figure 1 ijerph-22-00040-f001:**
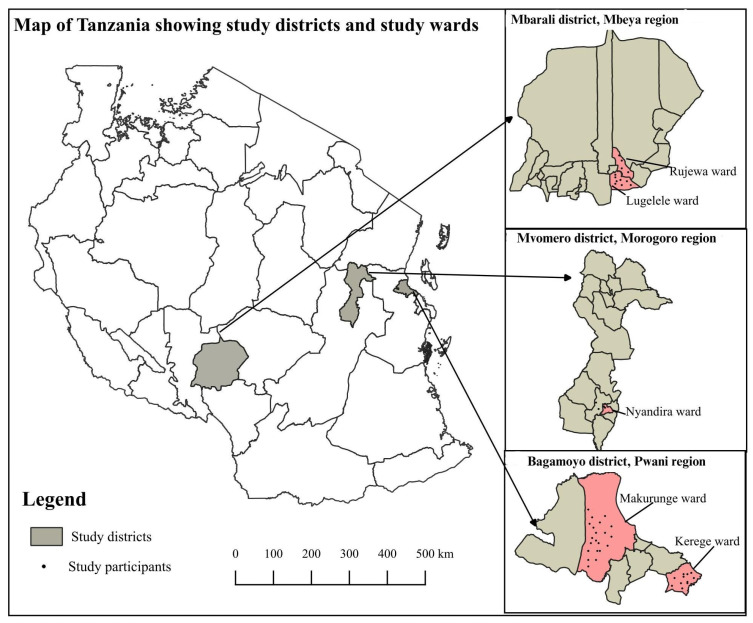
Study sites in Tanzania with districts and wards included (©W. N. Mwakalasya).

**Table 1 ijerph-22-00040-t001:** Social-demographic characteristics, time working in horticulture, and gestational age among the women from the three districts of Tanzania. The results from the three districts are compared (*n* = 432).


Variable	All (*n* = 432)	Mvomero (*n* = 258)	Bagamoyo (*n* = 80)	Mbarali (*n* = 94)	*p*-Value (χ^2^ Test)
	Frequency (%)	Frequency (%)	Frequency (%)	Frequency (%)	
Age					
≤30 years	141 (32.6)	67 (26.0)	28 (35.0)	46 (48.9)	
31–40 years	238 (55.1)	149 (57.7)	47 (58.8)	42 (44.7)	
Above 40 years	53 (12.3)	42 (16.3)	5 (6.2)	6 (6.4)	<0.001
Years in horticulture: *n* (%)					
≤5 years	239 (55.3)	135 (52.3)	57 (71.3)	47 (50.0)	
6–10 years	122 (28.2)	86 (33.3)	17 (21.3)	19 (20.2)	
Above 10 years	71 (16.4)	37 (14.3)	6 (7.5)	28 (29.8)	<0.001
Horticulture work during pregnancy: *n* (%)					
Yes	373 (86.3)	228 (88.4)	66 (82.5)	79 (84.0)	
No	59 (13.7)	30 (11.6)	14 (17.5)	15 (16.0)	0.313
Gestation age when stopping horticulture work					
First trimester	46 (10.6)	12 (4.7)	19 (23.8)	15 (16.0)	
Second trimester	150 (34.7)	71 (27.5)	31 (38.8)	48 (51.1)	
Third trimester	177 (41.0)	145 (56.2)	16 (20.0)	16 (17.0)	<0.001
Did not work	59 (13.7)	30 (11.6)	14 (17.5)	15 (16.0)	

**Table 2 ijerph-22-00040-t002:** Pesticide-contact activities practiced by pregnant women on horticulture farms in the three districts of Tanzania (the total number of participants: 373).

Pesticide-Contact Activity	Total (*n* = 373)	Mvomero (*n* = 228)	Bagamoyo (*n* = 66)	Mbarali (*n* = 79)	*p*-Value
	Frequency (%)	Frequency (%)	Frequency (%)	Frequency (%)	
Mixing of pesticides before spraying					
Practiced	31 (8.3)	11 (4.8)	12 (18.2)	8 (10.1)	
Did not practice	342 (91.7)	217 (95.2)	54 (81.8)	71 (91.5)	0.010 ^1^
Spraying of pesticides ^3^					
Practiced	146 (39.5)	84 (36.8)	30 (46.9)	32 (41.0)	
Did not practice	224 (60.5)	144 (63.2)	34 (53.1)	46 (59.0)	0.010 ^1^
Weeding within 24 h of spraying ^4^					
Practiced	217 (59.6)	113 (49.6)	48 (73.8)	56 (78.9)	
Did not practice	147 (40.4)	115 (50.4)	17 (26.2)	15 (21.1)	<0.001 ^1^
Harvesting within 24 h of spraying ^5^					
Practiced	85 (23.1)	16 (7.0)	33 (50.0)	36 (47.9)	
Did not practice	284 (77.0)	212 (93.0)	33 (50.0)	39 (52.1)	<0.001 ^1^
Washing clothes used in pesticide spraying ^3^					
Practiced	191 (51.3)	75 (32.9)	49 (74.2)	67 (85.9)	
Did not practice	181 (48.7)	153 (67.1)	17 (25.8)	11 (14.1)	<0.001 ^1^
Washing equipment used in spraying ^3^					
Practiced	106 (28.5)	8 (3.5)	35 (53.1)	63 (80.8)	
Did not practice	266 (71.5)	220 (96.5)	31 (46.9)	15 (19.2)	<0.001 ^1^
Burning pesticide container ^3^					
Practiced	9 (2.4)	0 (0.0)	4 (6.2)	5 (6.3)	
Did not practice	363 (97.6)	228 (100.0)	61 (93.8)	74 (93.7)	<0.001 ^2^
Reusing pesticide container					
Practiced	6 (1.6)	0 (0.0)	3 (4.5)	3 (3.8)	
Did not practice	367 (98.4)	228 (100.0)	63 (95.5)	76 (96.2)	0.004 ^2^
Eating farm products within 24 h of spraying ^6^					
Practiced	167 (45.8)	58 (25.4)	44 (67.7)	65 (90.3)	
Did not practice	198 (54.2)	170 (74.6)	21 (32.3)	7 (9.7)	<0.001 ^1^

^1^ Chi-square test; ^2^ Fisher’s exact test. ^3^ Missing 1 answer; ^4^ missing 9 answers; ^5^ missing 4 answers; ^6^ missing 8 answers.

**Table 3 ijerph-22-00040-t003:** Gestation age when pesticide-contact activities were performed on horticulture farms from the three districts in Tanzania (*n* = 373).

Gestation Age When Pesticide-Contact Activity Was Practiced	Total (*n* = 373)	Mvomero (*n* = 228)	Bagamoyo (*n* = 66)	Mbarali (*n* = 79)
	Frequency (%)	Frequency (%)	Frequency (%)	Frequency (%)
Mixing of pesticides before spraying				
First trimester	9 (2.4)	0 (0.0)	5 (7.6)	4 (5.1)
Second trimester	7 (1.9)	2 (0.9)	3 (4.5)	2 (2.5)
Third trimester	3 (0.8)	0 (0.0)	3 (4.5)	0 (0.0)
Spraying of pesticides				
First trimester	145 (38.9)	82 (36.0)	31 (47.0)	32 (40.5)
Second trimester	143 (38.3)	84 (36.8)	28 (42.4)	31 (39.2)
Third trimester	138 (37.0)	82 (36.0)	28 (42.4)	28 (35.4)
Weeding within 24 h of spraying				
First trimester	194 (52.0)	113 (49.6)	41 (62.1)	40 (50.6)
Second trimester	198 (53.1)	113 (49.6)	39 (59.1)	46 (58.2)
Third trimester	31 (8.3)	0 (0.0)	18 (27.3)	13 (16.5)
Harvesting within 24 h of spraying				
First trimester	40 (10.7)	2 (0.9)	19 (28.8)	19 (24.1)
Second trimester	68 (18.2)	14 (6.1)	24 (36.4)	30 (38.0)
Third trimester	52 (13.9)	11 (4.8)	28 (42.4)	13 (16.5)
Washing clothes used in pesticide spraying				
First trimester	175 (46.9)	75 (32.9)	42 (63.6)	58 (73.4)
Second trimester	70 (18.8)	0 (0.0)	20 (30.3)	50 (63.3)
Third trimester	66 (17.7)	0 (0.0)	23 (34.0)	43 (54.4)
Washing equipment used in spraying				
First trimester	95 (25.5)	8 (3.5)	29 (43.9)	58 (73.4)
Second trimester	96 (25.7)	8 (3.5)	29 (43.9)	59 (74.7)
Third trimester	18 (4.8)	0 (0.0)	7 (10.6)	11 (13.9)
Burning pesticide container				
First trimester	6 (1.6)	0 (0.0)	2 (3.0)	4 (5.1)
Second trimester	7 (1.9)	0 (0.0)	2 (3.0)	5 (6.3)
Third trimester	5 (1.3)	0 (0.0)	2 (3.0)	3 (3.8)

**Table 4 ijerph-22-00040-t004:** A comparison between the two groups of women in horticulture work based on their knowledge of pesticide handling (low or high) and the social-demographic variables (*n* = 432).

Variable	Low Knowledge	High Knowledge	*p*-Value
	Frequency (%)	Frequency (%)	
Aware of the information on the pesticide label			
Yes	272 (67.5)	29 (100)	
No	131 (32.5)	0 (0.0)	<0.001 ^2^
Knowledge	403 (93.3)	29 (6.7)	
Study area			
Mvomero	254 (63.0)	4 (13.8)	
Bagamoyo	56 (13.9)	24 (82.8)	
Mbarali	93 (23.1)	1 (3.4)	<0.001 ^1^
Mother’s age			
≤30 years	129 (32.0)	12 (41.4)	
31–40 years	224 (55.6)	14 (48.3)	
Above 40 years	50 (12.4)	3 (10.3)	0.558 ^1^
Years in horticulture			
≤5 years	230 (57.1)	15 (51.7)	
6–10 years	141 (35.0)	12 (41.4)	
>10 years	32 (7.9)	2 (6.8)	0.688 ^1^
Gestation age when stopping horticulture work			
First trimester	37 (9.2)	9 (31.0)	
Second trimester	138 (34.2)	12 (41.4)	
Third trimester	170 (42.2)	7 (24.1)	
Did not work	58 (14.4)	1 (3.4)	0.001 ^1^

^1^ Chi-square test; ^2^ Fisher’s exact test.

**Table 5 ijerph-22-00040-t005:** A multiple response frequency table showing the distribution of the women’s awareness of the contents of the pesticide label (*n* = 432).

Variable	Responses	Percentage
How to mix	270	62.5
How to spray	277	64.1
Personal protection	62	14.4
First aid	30	6.9
Toxicity	41	9.5
Storage	94	21.8
Disposal	25	5.8
Pesticide name	33	7.6

## Data Availability

Data are available on request.

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
