# Peer review of "Self-Reported Pesticide Exposure During Pregnancy and Pesticide-Handling Knowledge Among Small-Scale Horticulture Women Workers in Tanzania, a Descriptive Cross-Sectional Study"

_ijerph, 2024, doi:10.3390/ijerph22010040_

Round 1

Reviewer 1 Report

Comments and Suggestions for Authors

Abstract

1.    In the abstract, you mention, “This study suggests that women working in horticulture engage in activities that expose them to pesticides during pregnancy and have low knowledge of pesticide handling.” However, this sentence risks sounding as though these women are actively choosing to engage in hazardous activities, potentially overlooking the fact that many are likely exposed to these risks involuntarily. Please revise the sentence to make it more neutral and objective, reflecting that the risks stem from the working conditions rather than personal choices.

Materials and Methods Section

2.    Did you plot Figure 1 yourself? If not, you should include a citation below the map indicating the publisher, author, and publication year, or relevant details, even if you modified the map. If you are using a copyrighted map, make sure you have permission to use it, or choose maps with open licenses and follow the terms of those licenses.

3.    It seems more appropriate to summarize the questionnaire in the main text and move Table 1 to the appendix or supplementary material.

4.    It is problematic to code “I don’t remember” as “did not practice”, as they are not equivalent. A better approach would be to handle “Never” and “I don’t remember” separately. For example, code “I don’t remember” as a missing value or as a special category.

Results Section

5.    The main text usually summarizes the key findings, statistical significance, and explanations of relationships or trends when the table provides Chi-squared test results, but there is no explanation for the statistical results in Table 2 in the main text.

6.    For Table 2, it may be misleading to include the variables “Horticulture work during pregnancy” and “Gestation age when decided to stop horticulture work” in the same table as other variables due to differing sample sizes. One approach to consider is adding rows for “Horticulture work during pregnancy” with “Yes” and “No” categories, and adding a row for “Gestation age when decided to stop horticulture work” to include a “Do not work during pregnancy” category. By doing this, the percentages and Chi-squared tests can be properly calculated. The same issue applies to Table 5.
7.    For Table 3 and Table 4, you can use Fisher’s exact test instead of the Chi-squared test to handle zero cell counts.

8.    In Section 2.2, you mention, “Years in horticulture (experience) showed a significant association with eating farm products within 24 hours of post-spraying (p=0.044) only.” However, the p-value in the sentence is inconsistent with the result shown in Table 3.

9.    There are major issues with Table 4:
(1)    It appears that women in the later trimester are a subset of women in the former trimester. However, variables such as “Harvest within 24 hours post-spray” and “Burned pesticides container” do not reflect this trend, as the number of women in the 2nd trimester is larger than in the 1st trimester.
(2)    The percentages are calculated based on n = 432 instead of 373, so the caption is incorrect.
(3)    If women in the later trimester are indeed a subset of those in the former trimester, then the Fisher’s exact/Chi-squared test is invalid since the categories are dependent.

10.    In Section 2.2, you mention, “A chi-square test of independence showed that there was an association between study area and gestational age when women engaged in weeding within 24 hours after spray (p<0.001), spraying pesticides (p=0.002), harvesting within 24 hours after spray (p<0.001) and washing clothes used for pesticide spray (p<0.001) (Table 5).” However, I cannot see these results in Table 5.

11.    In Section 3.3, you mention, “When women were asked if they were aware of the presence of important pesticide information displayed on the packaging label, approximately 70% of them (n=301) reported being aware. Pearson’s chi-square test showed that awareness was significantly different among women from the three study districts, with the Mvomero district having the highest proportion of women who were aware.” However, I cannot see these results from anywhere in the paper. Please provide the detailed statistics.

12.    In Section 3.3, you mention, “The results of the Fisher’s exact test (p=0.862) do not indicate a significant association between years spent on horticulture (experience) and knowledge level”. However, the p-value of the Fisher’s exact test is inconsistent with the result shown in Table 5.

13.    Figure 2 is very confusing. There are several issues:
(1)    Please use a decimal point or percentage instead of a comma.
(2)    The sum of all percentages exceeds 100%.
(3)    If the percentages in the figure are correct, it means there is overlap among categories. In that case, it would not be appropriate to use doughnut chart.

Reviewer 2 Report

Comments and Suggestions for Authors

The manuscript "Self-reported Pesticide Exposure During Pregnancy and Pesticide Handling Knowledge Among Small-Scale Horticulture Women Workers in Tanzania" is well-organized and addresses a significant public health concern. However, specific observations and recommendations are made to improve its clarity, rigor, and conformity with publication standards.

The abstract offers a brief summary, but a greater focus on the findings concerning health hazards and policy consequences could enhance its effectiveness. The methods should be concisely outlined, and the introduction should clearly define the study's distinctiveness and significance within the Tanzanian setting. The objectives should be articulated more explicitly to public health outcomes, highlighting how this study can guide policy and actions.

The methodology should elaborate on the sampling method, the reliability and validity of the self-administered questionnaire, and the data examination. The material should be well-presented but could be improved by combining specific tables and highlighting regional disparities. The narrative should incorporate further analysis alongside the data to elucidate the ramifications of particular statistics.

The discussion should contextualize the findings with other research conducted in similar socioeconomic or agricultural settings and propose additional targeted recommendations for local policymakers, including plans for educational interventions and implementing safety requirements in horticulture. The limitations section should emphasize the dependence on self-reported data and the possibility of recollection bias.

In conclusion, the study's importance should be emphasized, and incorporating a targeted call to action, such as promoting gender-sensitive pesticide safety education, could enhance its efficacy.

Reviewer 3 Report

Comments and Suggestions for Authors

Dear Authors 

Many thanks for the chance to review your manuscript. Consider the following points to help improve the paper quality and appeal to your target audience 

i. The title seem to depict as though the participants were pregnant when the work was done. The title need to reflect the work done 

ii. Ensure your keywords are nit repetition of the paper title. Better you select operational words that will help improve the paper visibility 

iii. Lie 60-65, the paper present two sets of aims. Better if these are harmonized into a single aim for clarity. 

iv. Are the authors stating the data was extracted from participants in Mvomero ward? if so the preceding sentences will require modification (Line 70-76).

v. Section 2.1 is a bit wordy and recommend authors to rephrase the content and eliminate redundant words. 

vi. Table 1 should be reported as appendix instead considering the information was already reported in the text (Section 2.2.)

vii. Line 144-146 stated open ended questions was used however these results were not reported anywhere. This need revisiting

viii.  The result section will require modification to report on key results as contained in the respective table. 

ix. Better if the content in figure 2 is reported alongside related table . As it stand the content does not reflect the statement made in the related text (line 243-245)

x. Authors need to consider the study limitation 

xi. Overall, the paper will require further proofread to improve its content. 

Reviewer 4 Report

Comments and Suggestions for Authors

dear Authors, thank you for your work. it's very interesting for public health in a one health point of view. a question for you: why you didn't stratify women for their age? Did you consider whether the woman was in her first, second, etc. pregnancy?

Round 2

Reviewer 1 Report

Comments and Suggestions for Authors

Thank you for addressing my comments so thoroughly. Your revisions are great and have significantly enhanced the quality of the manuscript. Most of my comments have been addressed and corrected properly. I truly appreciate your effort and dedication. However, I have noted a few minor issues that remain after the revisions, which I have listed below.

Results Section

1.     The updated Table 1 (originally Table 2) is much clearer now. However, the percentages for the variable “Gestation age when decided to stop horticulture work” have not been corrected after including the “Did not work” category. Please ensure the percentages are corrected in both table and main text before publication. Additionally, please verify that the p-value for the Chi-squared test for this variable is also up to date. While it would not be surprising for the p-value to remain below 0.001, this is just a reminder.

2.     I see that Table 2 (originally Table 3) has been updated by removing missing values. Please include this information in the caption. For example, in parenthesis, you could write “n = 432 total, with missing values excluded per variable”. Additionally, please verify whether the percentages are accurate, as I noticed some of them differ from my calculations.

3.     I see that the caption of Table 3 (originally Table 4) has been corrected, but the table may still cause confusion regarding how the percentages are obtained. I would suggest adding the sample size at the top of the table, below the district names, as shown in Table 1.

In my opinion, the paper will be ready for publication once these revisions are made.

Author Response

Response to reviewer’s comments

Dear reviewers,

We would like to express our heartfelt gratitude to the editor and the reviewers for their meticulous review and constructive feedback on our manuscript. Your insightful comments and suggestions have been very important in improving the overall quality of our work. We have carefully addressed each point raised and made the necessary revisions to strengthen the manuscript. We greatly appreciate the time and effort you dedicated to this process, and we are confident that the improvements reflect your thoughtful guidance. The table in file enclosed.

Reviewer 3 Report

Comments and Suggestions for Authors

Dear Authors 

Many thanks for the time taken to improve the work further. While i could see great improvement made however there are few areas that need addressing to ensure the work appeal to its target audience: 

i. Table 4 variable "Aware of the information in the practice label" 272 said yes. When compared this with table 5 how to spray has 277 response which is higher than the value reported here. Please check this in my opinion should not exceed 272 reported earlier. 

ii. In addition, i will recommend further proof read to strengthen the work further  

Author Response

Response to reviewer’s comments

Dear reviewers,

We would like to express our heartfelt gratitude to the editor and the reviewers for their meticulous review and constructive feedback on our manuscript. Your insightful comments and suggestions have been very important in improving the overall quality of our work. We have carefully addressed each point raised and made the necessary revisions to strengthen the manuscript. We greatly appreciate the time and effort you dedicated to this process, and we are confident that the improvements reflect your thoughtful guidance. The table in the enclosed file presents point-by-point response to the reviewer’s comments.

Reviewer 4 Report

Comments and Suggestions for Authors

thank you for your work!

Author Response

Response to reviewer’s comments

Dear reviewer,

We would like to express our heartfelt gratitude to the editor and the reviewers for their meticulous review and constructive feedback on our manuscript. Thank you.